# The Prognostic Role of Pulmonary Arterial Elastance in Patients Undergoing Left Ventricular Assist Device Implantation: A Pilot Study

**DOI:** 10.3390/jcm13237102

**Published:** 2024-11-24

**Authors:** Marco Di Mauro, Michelle Kittleson, Giulio Cacioli, Vito Piazza, Rita Lucia Putini, Rita Gravino, Vincenzo Polizzi, Andrea Montalto, Marina Comisso, Fabio Sbaraglia, Emanuele Monda, Andrea Petraio, Marisa De Feo, Cristiano Amarelli, Claudio Marra, Francesco Musumeci, Emilio Di Lorenzo, Daniele Masarone

**Affiliations:** 1Azienda Sanitaria Locale Avellino (ASL AV), 83100 Avellino, Italy; 2Department of Cardiology, AORN dei Colli Monaldi Hospital, 80131 Naples, Italy; 3Division of Cardiology, Smidt Heart Institute, Cedars-Sinai Medical Center, Los Angeles, CA 90048, USA; 4Department of Cardiosciences, Division of Cardiology, Azienda Ospedaliera San Camillo-Forlanini, Circonvallazione Gianicolense 87, 00152 Rome, Italy; 5Division of Cardiology, V.Cervello Hospital, AOOR Villa Sofia -Cervello, 90146 Palermo, Italy; 6Department of Cardiac Surgery, Ospedale San Camillo, 00152 Rome, Italy; 7Department of Cardiac Surgery and Transplants, AORN dei Colli Monaldi Hospital, 80131 Naples, Italy

**Keywords:** advanced heart failure, left ventricular assist device, HeartMate 3, pulmonary arterial elastance

## Abstract

**Background**: Pulmonary arterial elastance (Ea) is a helpful parameter to predict the risk of acute postoperative right ventricular failure (RVF) after left ventricular assist device (LVAD) implantation. A new method for calculating Ea, obtained by the ratio between transpulmonary gradient and stroke volume (Ea^B^), has been proposed as a more accurate measure than the Ea obtained as the ratio between pulmonary artery systolic pressure and stroke volume (Ea^C^). However, the role of Ea^B^ in predicting acute RVF post-LVAD implantation remains unclear. **Methods and Results**: A total of 35 patients who underwent LVAD implantation from 2018 to 2021 were reviewed in this retrospective analysis. Acute RVF after LVAD implantation occurred in 12 patients (34%): 5 patients with moderate RVF (14% of total) and 7 patients with severe RVF. The Ea^B^ was not significantly different between the “severe RVF” vs. “not-severe RVF” groups (0.27 ± 0.04 vs 0.23 ± 0.1, *p* < 0.403). However, the combination of arterial elastance and central venous pressure was significantly different between the “not-severe RVF” group (central venous pressure < 14 mmHg and Ea^C^ < 0.88 mmHg/mL or Ea^B^ < 0.24 mmHg/mL; *p* < 0.005) and the “severe RVF” group (central venous pressure > 14 mmHg and Ea^C^ > 0.88 mmHg/mL or Ea^B^ > 0.24 mmHg/mL; *p* < 0.005). **Conclusions**: Ea is a reliable parameter of right ventricular afterload and helps discriminate the risk of acute RVF after LVAD implantation. The combined analysis of Ea and central venous pressure can also risk stratify patients undergoing LVAD implantation for the development of RVF.

## 1. Introduction

Advanced heart failure (advHF) represents the end-stage phase of heart failure, and it is characterized by a poor prognosis since limited therapeutic options are available [1]. In line with the standard clinical practice, if a patient experiences severe symptoms and frequent hospitalizations due to heart failure despite receiving guideline-directed medical therapy (GDMT) and is diagnosed with advanced heart failure with reduced ejection fraction (advHFrEF), options such as a heart transplant or the implantation of a left ventricular assist device (LVAD) should be contemplated [2]. The significant reduction in hemocompatibility-related adverse events with third-generation LVAD (HeartMate 3) has significantly improved the long-term outcome, with estimated survival rates of 83.4% at two years and 63.3% at five years [3]. However, right ventricular failure (RVF) may severely reduce these survival rates [4]. 

Particularly, acute RVF usually manifests within the first two weeks after LVAD implants with a low-flow syndrome that can lead to multi-organ dysfunction and a high risk of morbidity and mortality [5,6,7]. The mechanisms behind acute RVF are related to the hemodynamic effects of LVAD support; indeed, the LVAD unloads the left ventricle, causing a drop in left atrial end-diastolic pressure, with an increase in systemic flow and systemic venous return and, therefore, in right ventricular preload [8,9].

The prevalence of RVF is reported to vary significantly (9–40%) because of the varying definitions employed in registry studies [10,11,12]. Some studies exclusively defined acute RVF as requiring mechanical circulatory support for the right ventricle, whereas others encompassed cases that required the extended use of inotropes and vasodilators for more than 14 days [13,14]. 

According to the 2016 definition from the Interagency Registry for Mechanically Assisted Circulatory Support (INTERMACS), acute right ventricular failure (RVF) is characterized by signs of elevated central venous pressure (CVP) and clinical or laboratory indicators of congestion, categorized into four severity grades. Currently, there is no single clinical, echocardiographic, or hemodynamic parameter recognized as a dependable marker for assessing the risk of RVF following LVAD implantation [15,16,17,18,19,20,21,22,23]. Recently, pulmonary arterial elastance (Ea), which measures total right ventricular afterload, has been suggested as one of the most reliable predictors for post-LVAD RVF risk when compared to traditional parameters [24,25,26]. Typically, Ea is evaluated during right heart catheterizations as the ratio of systolic pulmonary artery pressure to stroke volume (SPAP/SV). Still, Brenner et al. [27] have recently introduced a different approach for assessing Ea using the following formula: (mean pulmonary arterial pressure - pulmonary capillary wedge pressure)/stroke volume. Since the predictive value of this innovative hemodynamic measure for post-LVAD RVF remains unclear, this study seeks to examine the prognostic significance of this new method of estimating Ea in predicting acute RVF following LVAD implantation.

## 2. Methods

In this prospective study, we enrolled all patients with a diagnosis of advHFrEF who underwent LVAD implantation between January 2018 and December 2021. 

The diagnosis of advHFrEF was performed according to European Society of Cardiology guidelines [28] by the presence of the following criteria despite guideline-directed medical therapy:(1)New York Heart Association class III (advanced) or IV;(2)Left ventricular ejection fraction (LVEF) ≤ 30%;(3)Persistence of a plasma level of NT-pro Brian Natriuretic Peptide > 2500 pg/mL;(4)Episodes of congestions (both pulmonary or systemic) requiring an IV dose of furosemide > 100 mg or the addition of metolazone or episodes of low output state requiring an infusion of inotropes or inodilators;(5)Distance covered at six-minutes walking distance test < 300 m or a peak VO_2_ < 12 mL/kg/min.

Electronic medical records were used to evaluate patients’ clinical and hemodynamic data.

The study was conducted according to the Declaration of Helsinki. 

All patients signed informed consent, and the Ethic Committee of AORN dei Colli–Ospedale Monaldi approved the study (deliberation number: 365 of December 2017). 

### 2.1. Echocardiography

Echocardiographic examinations were performed with a 3.5 MHz monoplane ultrasound probe from Vivid E-9 (GE Vingmed Ultrasound, Horten, Norway).

Two cardiologists for each center with expertise in cardiovascular imaging acquired and analyzed all pre-and post-LVAD implantation echocardiographic images. For each echocardiographic parameter, three cardiac cycles were measured in patients with sinus rhythm and five cardiac cycles in patients with atrial fibrillation. The mean value of this measurement was considered. 

As per international guidelines [29], LVEF was calculated by using the following formula: LVEF = [left ventricular end-diastolic volume (LVEDV) − left ventricular end-systolic volume (LVESV)]/LVEDV × 100. 

According to common practice, the tricuspid annulus plane systolic excursion (TAPSE) was also evaluated and assessed as the peak excursion of the tricuspid annulus (millimeters) from the end of diastole to end-systole. 

The inter-observer and intra-observer variability, which assessed the reproducibility of the measurements, was evaluated using Pearson’s two-tailed bivariate correlations and Bland–Altman analysis. The correlation coefficients, 95% confidence limits, and percentage errors were reported.

### 2.2. Right Heart Catheterization

Before LVAD implantation, all patients had right heart catheterization (RHC). An experienced cardiologist performed the RHC using a 6 French Swan–Ganz catheter (Edwards Lifesciences, Irvine, CAL, USA) which was inserted with fluoroscopic guidance through either the jugular or femoral vein. Catheter placement during the procedure was verified by fluoroscopy and by analyzing the pressure waveform characteristics. In line with the standard clinical practice [30], the manometer was zeroed at the level of the middle axillary line. The central venous pressure (CVP), systolic pulmonary arterial pressure (SPAP), pulmonary capillary wedge pressure (PCWP), and MPAP were recorded as an average of three beats. The thermodilution technique was utilized to calculate stroke volume (SV), cardiac output (CO), and cardiac index (CI).

### 2.3. Evaluation of Pulmonary Elastance

Ea was derived from hemodynamic data and calculated using two different methods. 

(1)The conventional method with the following formula: PASP/SV ratio (EaC);(2)The recent one presented by Brenner et al. with the following formula: (MPAP − PCWP)/SV ratio (EaB).

### 2.4. Left Ventricular Assist Device Implant

All patients were implanted with a centrifugal flow pump third-generation LVAD (HeartMate 3^®^ LVAD, Abbott, Chicago, IL, USA). 

The implantation of the HeartMate 3 was conducted by skilled cardiac surgeons specializing in the surgical management of advHFrEF. The procedure utilized standard median sternotomy along with cardiopulmonary bypass or extracorporeal membrane oxygenation, based on the surgeon’s choice. The apical cuff was stitched onto the epicardial surface near the left ventricular apex, and a myocardial nucleus was excised using a circular knife. The targeted area for myocardial coring was along the interventricular septum towards the mitral orifice. The pump was positioned at the left ventricular apex, with the outflow graft anastomosed to the ascending aorta following usual surgical protocols. The inflow cannula was then inserted into the left ventricle through the apex and connected to the apical cuff. Lastly, the driveline was tunneled and brought out through the abdominal wall.

### 2.5. Definition of RVF

The diagnosis of postoperative acute RVF was made according to INTERMACS adverse event definition criteria [31]: (1)Symptoms or signs of RVF;(2)CVP > 16 mmHg at RHC;(3)Clinical evidence of congestion (peripheral edema printable with grade 2 or more), ascites, hepatomegaly (on clinical examination or liver echography), and biochemical parameters of hepatic or renal distress.(4)Occurrence < 30 days after LVAD implant.

As illustrated in Figure 1, RVF was classified into mild, moderate, severe, or severe-acute categories. To enhance statistical power, the patient population was split into two groups: “severe” (including patients with severe and severe-acute RVF grades who needed extended post-implant inotropes, vasodilators, right ventricular assist device implantation, or died primarily due to acute RVF) and “not-severe’ (comprising mild and moderate RVF grades).

### 2.6. Statistical Analysis

Normally distributed continuous variables are described as mean ± standard deviation with two or three group comparisons conducted using Student *t*-test and analysis of variance (ANOVA), respectively. Skewed data are represented as median (interquartile range [IQR]) with two- or three-group comparisons performed using Wilcoxon rank-sum and Kruskal–Wallis tests, respectively. Categorical variables are listed as numbers (percentage) with group comparison conducted using χ^2^ test or Fisher’s exact test. A significance level (*p*-value) of 0.05 (two-sided test) was used for all the comparisons. All statistical analyses were performed using IBM SPSS Statistics for Macintosh, Version 27.0.

## 3. Results

The final cohort comprises 35 patients; of these, 23 patients (66%) had advHFrEF of ischemic etiology (67%), 10 patients (29%) presented idiopathic etiology of advHFrEF, and 2 patients (5%) had advHFrEF due to an end-stage form of hypertrophic cardiomyopathy. 

The clinical and echocardiographic characteristics of the patient population are summarized in Table 1.

A total of 19 implants (54%) were performed as destination therapy, while 16 implants (46%) served as bridge to transplant. Acute right ventricular failure (RVF) post-LVAD implantation was observed in twelve patients (34%), including five patients with moderate RVF (14% of the total), two patients with severe RVF (0.06%), and five patients with severe acute RVF that necessitated RVAD implantation (14%). Three individuals who received RVAD implantation were evaluated in advance as high risk for RVF and were provided temporary RVAD support during LVAD implantation, leading to their exclusion from later analysis. Four patients (12%) died during the postoperative phase. Among the nine hemodynamic parameters analyzed, none showed an independent correlation with the risk of RVF following LVAD (*p* < 0.06; Table 2). 

The pulmonary arterial elastance calculated using the conventional method (Ea^C^) was higher in the severe RVF group compared to the non-severe RVF group (1.1 ± 0.2 vs. 0.95 ± 0.4), although this difference lacked statistical significance (*p* = 0.361). The cut-off for Ea^C^ was established at 0.88 mmHg/mL, determined from ROC curves, with a specificity of 53% and sensitivity of 100%. The ROC curve identified a cut-off of 0.24 mmHg/mL for the elastance measured via the new method (Ea^B^) with specificity at 64% and sensitivity at 100% (Figure 2).

Using this cut-off value Ea^B^ showed a consistent difference between the “severe RVF” and “not severe RVF” groups, although it did not achieve statistical significance (0.28 ± 0.04 vs. 0.23 ± 0.1, *p* = 0.403; Figure 3). 

A combined analysis of arterial elastance and CVP aimed to evaluate potential right ventricular adaptation to increased afterload. In the “not severe RVF” group, the predominant hemodynamic status during preoperative catheterization showed CVP < 14 mmHg and EaC < 0.88 mmHg/mL (or EaB < 0.24 mmHg/mL), which was absent in patients with severe RVF. Conversely, the severe RVF group displayed a preoperative hemodynamic profile marked by CVP > 14 mmHg and EaC > 0.88 mmHg/mL (or EaB > 0.24 mmHg/mL). The disparities between the two groups were statistically significant for both arterial elastance calculation methods (*p* < 0.005; Table 3).

## 4. Discussion

In the past decade, LVAD therapy has emerged as a viable option for patients with advanced heart failure with reduced ejection fraction (advHFrEF) who cannot undergo heart transplants, serving as either a destination therapy (for patients with absolute contraindication to heart transplant) or a bridging option for those with relative and transient contraindications to transplantation [32].

However, numerous patients experience RVF after LVAD implantation, resulting in increased rates of heart failure-related hospitalizations and higher mortality compared to those without post-LVAD right ventricular failure (RVF) [33]. Consequently, RVF development poses a significant complication for patients on long-term LVAD support, restricting the overall effectiveness of the device therapy [34]. Significant progress has been made in recent years to discover prognostic markers for RVF after LVAD implantation. Several echocardiographic, biochemical, and hemodynamic indices have been identified as variable reliability markers for post-LVAD RVF; however, no single prognostic marker exists for predicting RVF onset following LVAD implantation [35,36,37].

Given the heightened mortality associated with RVF onset in LVAD patients, clarifying the hemodynamic parameters predictive of RVF (especially severe RVF) is critical. Recent studies have identified Ea, which reflects the afterload effects on the right ventricle [38,39], as a more dependable predictor of adverse outcomes compared to other hemodynamic parameters (like RVP and PCWP) in patients with pulmonary hypertension due to left heart disease (group 2 pulmonary hypertension) [40].

One previous study involving 375 patients with advHFrEF undergoing LVAD implantation revealed that an Ea^C^ value > 1.16 could identify patients at risk for severe acute RVF (*p* = 0.02). Additionally, combining Ea^C^ with right atrial pressure may be the most effective hemodynamic parameter for pinpointing patients at risk of severe acute RHF (*p* < 0.001) [41].

In our study, contrasting with the aforementioned study’s findings, we found that Ea^B^ does not predict the risk of acute RVF following LVAD placement. Nevertheless, the combination of CVP and Ea^B^ significantly enhanced the prognostic value. Although a dysfunctional right ventricle can elevate CVP, various unrelated factors may also contribute to increased CVP [42]. Therefore, the association between CVP and Ea assists in identifying patients specifically exhibiting RV dysfunction due to increased afterload, which represents a subgroup of advHFrEF at higher risk of post-LVAD RVF [43]. In addition, we determined that patients with low CVP and low Ea had a low risk of severe RVF. Conversely, RV dysfunction and ventricular–arterial uncoupling associated with high CVP and high Ea significantly raised the RVF risk after implantation, achieving statistical significance. Our study revealed no notable difference between the two groups (“severe” RVF vs. “not-severe” RVF) for intermediate hemodynamic profiles defined by high CVP and low Ea or low CVP and high Ea, possibly due to selection bias. However, careful hemodynamic evaluation is essential for these patients, as they present a compromised hemodynamic state that may affect postoperative outcomes due to altered ventricular-arterial coupling. It is crucial to note that traditional echocardiographic parameters for assessing right ventricular systolic function (e.g., TAPSE and Sv wave), typically used for evaluating patients’ candidacy for extended mechanical circulatory support [44], do not correlate with post-implantation RVF risk due to their high load dependency. This indicates the importance of invasive hemodynamic assessments for all advHFrEF patients considered for long-term mechanical circulatory support [45].

## 5. Limitations 

This study has several limitations that affect the interpretation of the results. First, the number of patients from a single country is limited. Second, the inclusion of patients from two centers may increase the inter-operator variability of echocardiographic and hemodynamic parameters. In addition, the multiple aspects of LVAD implantation and right ventricular performance may make it impossible to control for each variable, making it difficult to analyze all relevant factors, which may further complicate preoperative RVF prediction. Furthermore, numerous factors, including perioperative and postoperative considerations, may influence right ventricular function after left LVAD implantation. This multifactorial nature makes it difficult to isolate specific predictors of RVF and may limit the predictive accuracy of any preoperative assessment. Finally, cross-sectional studies are limited in their ability to document changes over time. Despite these limitations, we believe that the results of our pilot study may pave the way for studies with larger populations and longer follow-up to confirm these preliminary findings.

## 6. Conclusions

In this study, we observed that Ea^B^ alone does not help to identify patients with acute RVF after LVAD implantation. To date, no single factor consistently discriminates those at increased risk of RVF after LVAD implantation, and multi-parametric assessment is likely to be essential. As RV afterload and ventricular–arterial coupling predict the residual contractile force of the ventricle and the possible response to hemodynamic changes, a combined analysis of Ea and CVP may stratify patients undergoing LVAD implantation for the risk of acute RVF.

## Figures and Tables

**Figure 1 jcm-13-07102-f001:**
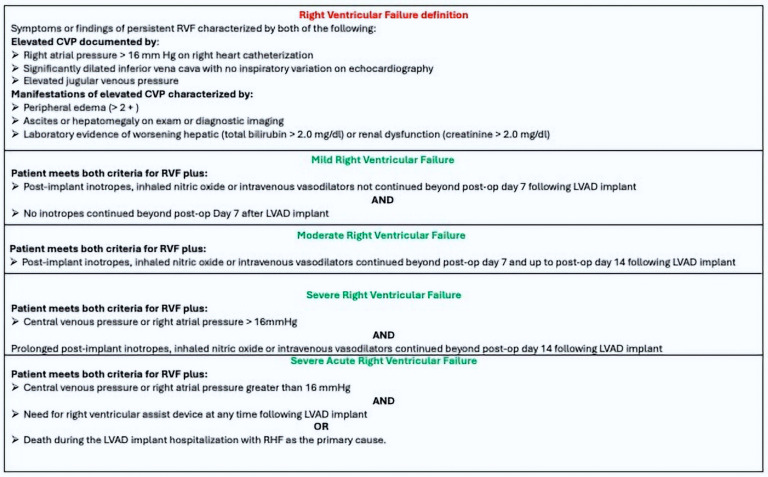
Definition and grading of right ventricular failure post-left ventricular assist device implant according to INTERMACS classification. CVP: central venous pressure; RVF: right ventricular failure; LVAD: left ventricular assist device.

**Figure 2 jcm-13-07102-f002:**
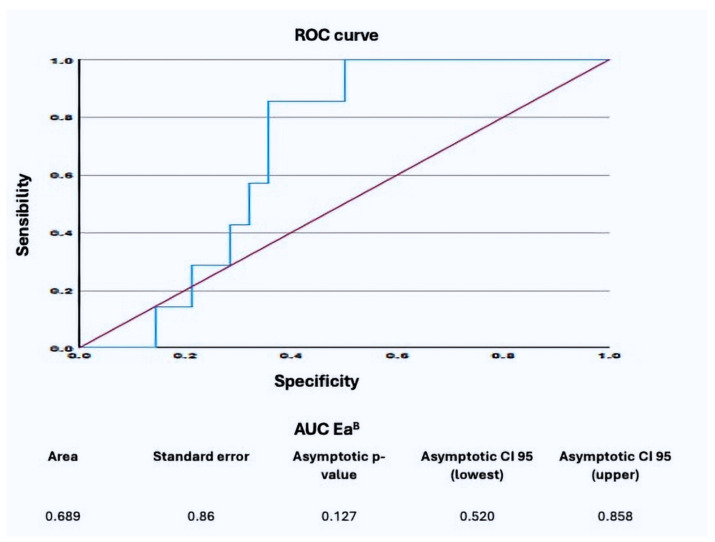
Receiver operating characteristic (ROC) curve of Ea^B^.

**Figure 3 jcm-13-07102-f003:**
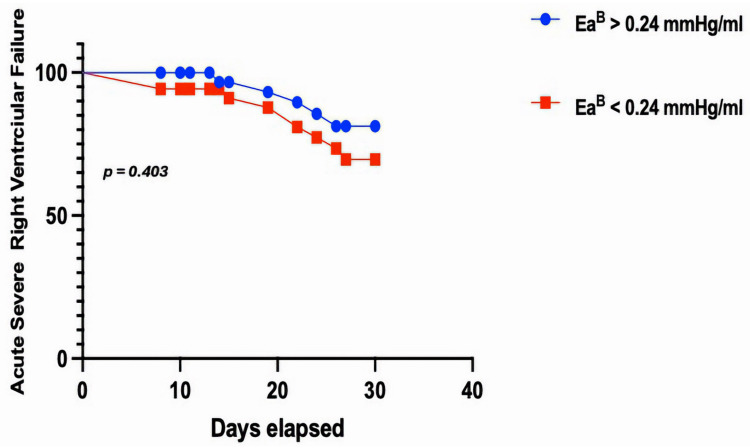
Kaplan–Meyers curve of acute severe right ventricular failure according to Ea^B^.

**Table 1 jcm-13-07102-t001:** Demographics and clinical characteristics of study population.

	All
Patients	35
Age	59 ± 12
Sex	33 M, 2 F
BMI	26 ± 3 kg/m^2^
LVEF	25 ± 6%
TAPSE	17.4 ± 2.8 mm
Ischemic etiology	23 (66%)
Idiopathic dilated cardiomyopathy	10 (29%)
End-stage hypertrophic cardiomyopathy	2 (5%)
HeartMate 3	35 (100%)

**Table 2 jcm-13-07102-t002:** Univariate analysis of echocardiographic and invasive hemodynamic variables. TAPSE: tricuspid annular plane excursion; TPG: transpulmonary pressure gradient; DPG: diastolic pressure gradient; PVR: pulmonary vascular resistance; RVSWi: right ventricular stroke work index; CPI: cardiac power index; Ea^C^: pulmonary arterial elastance (PASP/SV ratio); Ea^B^: pulmonary arterial elastance (MPAP-PCWP)/SV ratio).

Parameter	Not-Severe RVF (*n* = 28)	Severe RVF (*n* = 6)	*p*-Value
TAPSE	17.12 ± 3.9	17.8 ± 2.13	0.672
TPG	13.4 ± 9.1	13.8 ± 3.3	0.916
DPG	5.37 ± 8.34	2.66 ± 1.5	0.405
PVR (WU)	3.44 ± 2.57	4.36 ± 1.17	0.441
RVSWi	973.3 ± 714.99	934.3 ± 312.69	0.897
CPI	0.18 ± 0.053	0.177 ± 0.046	0.875
Compliance	3.055 ± 1.68	1.833 ± 0.401	0.09
Ea^C^	0.947 ± 0.43	1.11 ± 0.2	0.361
Ea^B^	0.232 ± 0.13	0.278 ± 0.04	0.403

**Table 3 jcm-13-07102-t003:** Multi-parametric analysis of invasive hemodynamics variables. CVP: central venous pressure; Ea^C^: pulmonary arterial elastance (PASP/SV ratio); Ea^B^: pulmonary arterial elastance (MPAP-PCWP)/SV ratio).

	Severe RVF	Not-Severe RVF		Severe RVF	Not-Severe RVF
CVP < 14 mmHg; Ea^C^ < 0.88 mmHg/ml	0%	46.4%	CVP < 14 mmHg; Ea^B^ < 0.24 mmHg/ml	0%	57.14%
CVP < 14 mmHg; Ea^C^ ≥ 0.88 mmHg/ml	66.6%	46.4%	CVP < 14 mmHg; Ea^B^ ≥ 0.24 mmHg/ml	66.6%	35.7%
CVP ≥ 14 mmHg; Ea^C^ < 0.88 mmHg/ml	0%	7.1%	CVP ≥ 14 mmHg; Ea^B^ < 0.24 mmHg/ml	0%	7.1%
CVP ≥ 14 mmHg; Ea^C^ ≥ 0.88 mmHg/ml	33%	0%	CVP ≥ 14 mmHg; Ea^B^ ≥ 0.24 mmHg/ml	33%	0%

## Data Availability

The data presented in this study are available on request from the corresponding author due to privacy reasons.

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
