# Peer review of "The Prognostic Role of Pulmonary Arterial Elastance in Patients Undergoing Left Ventricular Assist Device Implantation: A Pilot Study"

_jcm, 2024, doi:10.3390/jcm13237102_

Round 1
Reviewer 1 Report
Comments and Suggestions for Authors
The ideea of the study is a very interesting one with large clinical applications in the future, starting from the ideea that Pulmonary arterial elastance is a helpful parameter to predict the risk of acute postoperative right ventricular failure after left ventricular assist device implantation.
The study is well done, the design is clear .
but the figure 1 seems to be copied, it has no legend, no references
I suggest to be rewritten or reformulated
Comments on the Quality of English Languageminor english revisions
Author Response
The idea of the study is a very interesting one with large clinical applications in the future, starting from the idea that Pulmonary arterial elastance is a helpful parameter to predict the risk of acute postoperative right ventricular failure after left ventricular assist device implantation.
The study is well done, the design is clear .
but the figure 1 seems to be copied, it has no legend, no references
I suggest to be rewritten or reformulated
Response: Thanks to the reviewer for the positive comments on our work. We have modified Figure 1 according to the suggestion
Reviewer 2 Report
Comments and Suggestions for Authors
Interesting article
It is true that these are super-selected patients, but the cases are too few to draw conclusions
The topic of right ventricular dysfunction is still controversial
Can we have a ROC curve of the predictive capacity of Ea?
Can we have a temporal data on right ventricular dysfunction in a KM curve?
Author Response
Interesting article
It is true that these are super-selected patients, but the cases are too few to draw conclusions
The topic of right ventricular dysfunction is still controversial
Can we have a ROC curve of the predictive capacity of Ea?
Can we have a temporal data on right ventricular dysfunction in a KM curve?
Response: We agree with the reviewer that the topic of RV after LVAD implant is controversial. According to the suggestions, we have added a ROC curve for EaB and a KM curve
Reviewer 3 Report
Comments and Suggestions for Authors
Comments to the Author:
This manuscript by Mauro and colleagues entitled " Prognostic role of pulmonary arterial elastance in patients undergoing LVAD implantation: A pilot study. This study evaluated 35 LVAD patients from 2018 to 2021 to assess the predictive value of pulmonary arterial elastance (Ea) for acute RVF. While Ea calculated by the transpulmonary gradient to stroke volume (EaB) wasn’t significantly different between severe and non-severe RVF groups, a combination of Ea and central venous pressure successfully stratified risk for acute RVF. While EaB alone may not effectively predict severe RVF, the use of both metrics offers a promising approach to assess and manage RVF risk in this population. Although it shows some interesting results, several points regarding the experimental design need to be addressed. The following major suggestions are for the authors and editors' considerations. Some specific comments are as follows:
Major:
1. The study is characterized by a limited geographical scope, as it probably centered on patients from a singular location or region. This could hinder the applicability of the results to other demographic groups or healthcare settings.
2. The result may not adequately reflect long-term results or variations in right ventricular function over time, which are essential for comprehending the complete effects of left ventricular assist device implantation.
3. The multifaceted aspects of LVAD implantation and right ventricular performance may render it impossible to control for every variable, thereby complicating the analysis of all relevant factors.
4. The study acknowledges that numerous factors, including perioperative and postoperative considerations, can influence right ventricular (RV) function after left ventricular assist device (LVAD) implantation. This multifactorial nature makes it challenging to isolate the specific predictors of right ventricular failure (RVF) and may limit the predictive accuracy of any preoperative assessment.
5. Deficiency in longitudinal data: Cross-sectional studies are limited in their ability to document changes over time.
Comments on the Quality of English Language
Minor editing of English language required.
Author Response
This manuscript by Mauro and colleagues entitled " Prognostic role of pulmonary arterial elastance in patients undergoing LVAD implantation: A pilot study. This study evaluated 35 LVAD patients from 2018 to 2021 to assess the predictive value of pulmonary arterial elastance (Ea) for acute RVF. While Ea calculated by the transpulmonary gradient to stroke volume (EaB) wasn’t significantly different between severe and non-severe RVF groups, a combination of Ea and central venous pressure successfully stratified risk for acute RVF. While EaB alone may not effectively predict severe RVF, the use of both metrics offers a promising approach to assess and manage RVF risk in this population. Although it shows some interesting results, several points regarding the experimental design need to be addressed. The following major suggestions are for the authors and editors' considerations. Some specific comments are as follows:
Major:
- The study is characterized by a limited geographical scope, as it probably centered on patients from a singular location or region. This could hinder the applicability of the results to other demographic groups or healthcare settings.
Response: The study is conducted in two centers in South Italy; however, we have compared our study population to patients who underwent LVAD implant in 2024 at Cedar Sinai (CA; USA) thanks to M.K., who is a co-author. No difference was found between the two populations so we think that our results can be generalized to all patients with advHF
- The multifaceted aspects of LVAD implantation and right ventricular performance may render it impossible to control for every variable, thereby complicating the analysis of all relevant factors.
Response: We agree with this comment; however, we think that the results of our pilot study could lead to studies with a larger population and a longer follow-up.
- The study acknowledges that numerous factors, including perioperative and postoperative considerations, can influence right ventricular (RV) function after left ventricular assist device (LVAD) implantation. This multifactorial nature makes it challenging to isolate the specific predictors of right ventricular failure (RVF) and may limit the predictive accuracy of any preoperative assessment.
Response: We agree with this comment, which can be applied to all studies conducted in this matter; however, we think that the results of our pilot study could pave the way for studies with a larger population and a longer follow-up.
- Deficiency in longitudinal data: Cross-sectional studies are limited in their ability to document changes over time
Response: We agree with this comment, which can be applied to all studies conducted in this matter; however, we think that the results of our pilot study could pave the way for studies with a larger population and a longer follow-up.
Round 2
Reviewer 1 Report
Comments and Suggestions for Authors
The new version of the article was improved but I suggest to replace figure nr 1 with an original version.
Comments on the Quality of English Language
minor revisions
Author Response
The new version of the article was improved but I suggest to replace figure nr 1 with an original version
Thank you to the reviewer for their appreciation of our work
We have modified Figure 1 according to the suggestion